# Optimizing breeding strategies for early-maturing white maize through genetic diversity and population structure

**Hellen Mawia Mukiti**[1,2], **Baffour Badu-Apraku**[2], **Ayodeji Abe**[3], **Idris Ishola Adejumobi**[2*], **John Derera**[2]

**1** Pan African University Life and Earth Sciences Institute (including Health and Agriculture), University of Ibadan, Ibadan, Oyo State, Nigeria, **2** International Institute of Tropical Agriculture (IITA) Ibadan, Ibadan, Nigeria, **3** Department of Crop and Horticultural Sciences, University of Ibadan, Ibadan, Oyo State, Nigeria

* i.adejumobi@cgiar.org

## Abstract

Maize production and productivity in sub-Saharan Africa are constrained by various factors. Assessing the genetic diversity of newly developed elite inbred lines can help identify lines with desirable genes and explore genetic relatedness for heterotic breeding. The objectives of this study were to assess the level of genetic diversity, and population structure, and identify appropriate clustering methods for assigning maize inbreds into heterotic groups. Three hundred and seventy-six elite inbreds extracted from three source populations were genotyped using Diversity Array Technology (DArTtag) mid-density platform. Results from 1904 of 3,305 SNP marker obtained revealed average marker polymorphism information content (PIC) of 0.39, observed heterozygosity of 0.02, gene diversity of 0.37, minor allele frequency of 0.29, Shannon and Simpson indices of 6.86 and 949.09, respectively, and allele richness of 787.70. The optimum sub-population was three defined by an admixture-based model and principal component analysis. The average genetic distance was 0.303 varying from 0.03 (TZEI 2772 × TZEI 2761) to 0.372 (TZEI 2273 × TZEI 2832). For appropriate heterotic classification of the 376 elite inbreds, the use of IBS distance matrix and average linkage clustering method provided the highest cophenetic correlation coefficient (0.97). Three heterotic group (HG) were identified using IBS distance and average linkage clustering method with HG 1 have 188 inbreds, HG 2 having 137, and HG 3 having 59 inbreds. The pedigree-based phylogenetic tree showed substantial consistency with the heterotic groups identified. The F-statistics based on the underlying population structure revealed 10% variation among sub-populations and 90% variation within sub-populations with a moderate level of genetic differentiation (0.10). The elite inbred lines showed a high degree of genetic diversity, which could be beneficial for developing new, early-maturing white hybrids to mitigate production constraints in sub-Saharan Africa.

## Introduction

Maize (*Zea mays* L.) is globally cultivated for several purposes including food, feed, and industrial needs [1]. In sub-Saharan Africa (SSA), it is a primary food source and exhibits

**Data availability statement:** All relevant data are within the paper and its Supporting information files.

**Funding:** Bill and Melinda Gates Foundation through the Accelerated Genetic Gains and Stress Tolerant Maize for Africa Projects [OPP1134248]. - The role of the funder "Bill and Melinda Gates Foundation" was clarified as the payer of the manuscript publication fees in the manuscript.

**Competing interests:** The authors have declared that no competing interests exist.

the greatest capacity for crop production in the savanna of the sub-region. Maize accounts for 40% of the African cereal production with utilization extending from direct consumption as food by the populace to processed products and feeds for animals [2]. Maize consists of about 72% carbohydrate, 10% protein, and 4% fat, with an energy density of 365 Kcal/100 g [2]. Approximately 60% of the populace in SSA depend on agriculture as their primary source of sustenance and income [3]. However, due to the low productivity of maize constantly observed on farmers' fields, its potential to enhance food security is far from being met at the household and national levels. This is attributable to several constraints, which include drought, heat, low soil nitrogen, *Striga* parasitism, fall armyworm infestation, and several diseases [4]. Boosting maize production against these stress conditions is crucial to ensuring food security and livelihoods in SSA [5].

Genetic diversity is the raw material for any crop improvement program. Knowledge of the level of genetic diversity in germplasm is essential as it enables efficient allocation of resources to effectively meet breeding goals. The use of genetic variability in maize has been employed to select genetically diverse parents for hybrid production. Maize, being a species that experiences out-crossing, possesses a genome that is intricate and exhibits a substantial level of genetic diversity [6]. These characteristics are beneficial to breeders as it allows them to attain a considerable increase in hybrid vigour, resulting in the development of maize varieties with high productivity. Precise data on the genetic diversity of maize lines in a breeding program could facilitate the identification and classification of inbreds into heterotic groups. This would allow the development of productive hybrids from contrasting heterotic groups without the need to test every possible combination among prospective parents [7]. In assessing genetic diversity in maize germplasm, several methods including morphological, biochemical, and molecular markers have been employed. Of these methods, molecular markers provide more accurate and reliable results as it is independent of environmental influence [8]. Utilizing marker-based genetic distance (GD) estimates could help to avoid making hybrid crosses between closely related inbreds, especially when the heterotic groups of these lines are not clearly defined. Predictive analysis could be used to eliminate crosses with unsatisfactory performance and reduce the cost and the time involved in the identification of desired hybrids [9]. Over the decades, PCR-based markers, such as random amplified polymorphic DNA (RAPD), amplified fragment length polymorphism (AFLP), simple sequence repeats (SSR), and single nucleotide polymorphism (SNP) markers have demonstrated their efficacy as robust techniques for analyzing genetic variation and characterizing population structure in maize lines [7]. However, SNP markers have gained high popularity in various genetic studies of the modern breeding era owing to their advantages over other marker types [10].

In the last four decades, maize breeders at the International Institute of Tropical Agriculture (IITA; https://www.iita.org/) and International Maize and Wheat Improvement Center (CIMMYT; https://www.cimmyt.org/) in collaboration with the National Agricultural Research System (NARS) of several maize-growing countries have developed diverse range of germplasms. These germplasms comprised elite inbreds, populations, synthetics, and open-pollinated varieties and more recently hybrids [4]. Numerous public and private organizations use IITA and CIMMYT maize germplasm for a variety of purposes, including the development of open-pollinated varieties, production of hybrid seeds, breeding based on pedigree information, population development for mapping quantitative trait loci (QTL), production of doubled haploids, and the incorporation of traits using transgenic methods [11]. Improving food security in SSA depends on the development and deployment of superior maize genotypes preferably hybrids that combine tolerance to the stress factors characterizing SSA with high-yielding capability, to replace the genetically weak varieties cultivated

by many farmers. Hybrid production is of major interest owing to their superiority in grain yield from heterosis [12]. "Heterosis", a phenomenon elucidating the superiority of a hybrid over the mid-parent, better parent, or a reference genotype performance is only possible if there are distinct heterotic groups. Therefore, developing hybrid maize genotypes to boost maize production and productivity in SSA requires identifying parental inbreds from complementary genetic backgrounds and showing adequate levels of genetic diversity. Source populations with a high genetic variability could facilitate the extraction of such valuable inbreds [11].

In defining heterotic groups, adequate measures need to be taken to identify the appropriate genetic distance matrix and linkage method to summarize the best dissimilarity matrices into distinct genetic groups or clusters [13]. Hierarchical clustering is the most widely used approach in the analysis of crop diversity and heterotic classification due to its simplicity, interpretability, and ability to reveal underlying genetic relationships [14]. Popularly used genetic distance matrices include Identical-by-State (IBS), Gower distance (GD), Euclidean distance (Roger), and Jaccard distance and for linkage methods include complete linkage, single linkage, average linkage (UPGMA), centroid, ward.D2, McQuitty, and Median [13]. Each of these approaches has some distinctive features and may generate different results. Different distance and clustering methods have their strengths and weaknesses. While IBS is simple, it's sensitive to allele frequencies. Gower distance handles mixed data but can be affected by variable numbers. Roger's distance is geometrically intuitive but assumes linearity. Jaccard distance is suitable for binary data but sensitive to rare alleles [13]. In terms of clustering methods, single linkage can produce chaining effects, complete linkage can be overly conservative, average linkage can be sensitive to noise, and Ward's, McQuitty, and Median methods can be computationally intensive and sensitive to outliers [13]. It therefore becomes imperative to identify the most appropriate combination of distance matrix and linkage method for the assignment of the inbred lines into heterotic groups. The objectives of this study were to assess the level of genetic diversity and population structure of 376 early white maize inbred lines, and to identify the best distance matrix and clustering linkage methods for assigning the inbreds into distinct heterotic groups.

## Materials and methods

### Genetic materials

Three hundred and seventy-six elite $S_7$ early white multiple stress tolerant inbred lines were used for this study (S1 Table). The inbred lines were newly developed from three source populations namely: DTE STR-W Syn Pop C4, TZE-W Pop DT C5 STR C5, and TZEI 65 x ENT 11. Badu-Apraku and Fakorede [4] documented a detailed report on the development of the 367 elite inbreds from these source populations. Briefly, DTE STR-W Syn Pop C4 was developed from the broad-based population DTE STR-W Syn Pop following four cycles of $S_1$ family recurrent selection. TZE-W Pop DT C5 STR C5 was developed from the broad-based population TZE-W Pop DT STR following five cycles of $S_1$ family recurrent selection. The biparental population TZEI 65 x ENT 11 was developed from the parental inbreds from IITA (TZEI 65) and CIMMYT (ENT 11). The development of the inbreds from these three source populations began with the generation of $S_1$ lines. These lines were advanced through $S_2$ to $S_4$ stages of inbreeding by selfing. After each cycle of selfing, the lines were evaluated under artificial *Striga* infestation and induced moisture stress. At $S_4$, 250–300 selected lines were crossed to a broad-based tester to estimate their general combining ability effects and based on the testcrosses performance, 90–100 $S_4$ lines were advanced through $S_5$ to $S_7$ stage of inbreeding where homozygosity of the lines is expected to be 99%.

## Leaf sampling and genotyping

Seeds of the S7 inbreds were planted in the field and leaf samples were collected at three weeks after planting. The samples were kept under −80 degrees Celsius conditions for 72 hours and lyophilized using a Labconco Freezone 2.5 L System lyophilizer (Marshall Scientific, USA). Four leaf discs from the lyophilized leaf samples of each inbred were punched, placed in deep-well plate, and shipped to Intertek Laboratory in Australia, for mid-density DArTtag genotyping. The genotyping involved a series of steps, which combined molecular biology techniques with bioinformatics analysis described by Kilian et al. [15]. To begin with, DNA was isolated from the leaf samples, purified, and digested with specific restriction enzymes that cut it at predetermined target sequences. These target sequences were taken through adapter ligation during which adapters containing unique oligonucleotide sequences were ligated to the ends of the restriction fragments. Following adapter ligation was representation by complexity reduction through a proprietary amplification technique that selectively enriched specific genomic regions containing the pre-selected SNP markers. The PCR amplification followed to generate millions of copies of the target sequence used for hybridization and single nucleotide polymorphism (SNP) analysis. This allowed the detection of variations at the targeted SNP sites. The hybridized probes were taken for imaging and data analysis on the DArT platform to identify the presence or absence of specific polymorphisms at each SNP marker locus for each sample. After this, data quality control and genotyping calls were done [15].

## Marker quality control

Multiple sequences were generated by the DArTtag platform using proprietary analytical pipelines. The sequence results received from the DArT platform had 3,305 SNP markers. From this, quality control was performed with SNP QC implemented in R using minor allele frequency (MAF) (0.05), call rate ($\geq 90$), missing rate (0.2), genotype quality (GQ = 20), maximum and minimum allele = 2, and no indels. The markers were reduced to 1904 SNP SNPs after the filtering process and were used for downstream analyses.

## Assessment of genetic diversity, population structure, and heterotic pattern quality

The final HapMap with 1904 high-quality SNP data was first converted to variant call format (VCF). Genetic diversity indices such as expected and observed heterozygosities (EH and OH), minor allele frequency (MAF), and polymorphism information content (PIC) that are crucial tools for assessing the genetic variation within and among populations were estimated using the VCF file in PLINK 2.0 [16]. Shannon and inverse Simpson diversity as well as Alpha index [17] were also estimated. These parameters (Shannon, inverse Simpson, and Alpha indices) provide quantitative measures of the richness and evenness of alleles in the study population. The HapMap file was also converted to a minor allele frequency format from where the gene content file was generated in R [18]. To reveal the genetic stratification in the elite inbred population, the gene content file containing the 1904 DArT markers was imported into the Bayesian Markov chain Monte Carlo (MCMC) software STRUCTURE V2.3.4 [19]. Structure simulations were carried out using a burn-in period of 20,000 iterations and a MCMC set at 20,000 with no prior information on the origin of individuals. For the appropriate K-value, the Evanno transformation method was used which exhibited a low cross-validation error compared to other K values [20] The results obtained from STRUCTURE were uploaded into Harvester to determine the optimum K-value. A membership probability level ≥ 80 was used to assign the inbreds to sub-populations, while inbreds with a probability level < 80 were designated as admixture. Additionally, principal component analysis (PCA) was used to visualize

the pattern of the genetic structure and the relationship among the 376 inbred source populations using the FactorMineR package [21] in R. To identify the best distance matrix and linkage methods for assigning the early white inbreds into heterotic groups, four different genetic distance matrices were generated from the gene content file namely identity-by-state (IBS), Gower distance, Euclidian distance, and Jaccard distance based on Eqns 1–4, respectively.

$$\text{IBS}(i,j) = \frac{\text{number of alleles shared between individuals i and j}}{\text{total number of alleles compared}} \tag{1}$$

$$d(i,j) = 1 - \frac{\left(\Sigma\_\{k=1\}^\{p\}S\_k(i,j)\right)}{p} \tag{2}$$

*Where; d(i, j) is the Gower distance between objects i and j; p is the total number of variables; S_k(i, j) is the similarity coefficient between objects i and j for variable k.*

$$d(p,q) = \text{sqrt}\left((p1-q1)^2 + (p2-q2)^2 + ... + (pn-qn)^2\right) \tag{3}$$

*Where; d(p, q) is the Euclidean distance between points p and q; p1, p2,..., pn are the coordinates of point p; q1, q2,..., qn are the coordinates of point q.*

$$\text{Jaccard}(A,B) = \frac{|A \cap B|}{|A \cup B|} \tag{4}$$

*Where; | A ∩ B | is the cardinality of the intersection of sets A and B (i.e., the number of elements common to both sets); | A ∪ B | is the cardinality of the union of sets A and B (the total number of elements in both sets, including duplicates).*

These matrices were combined with seven types of linkage methods namely Complete linkage, Single linkage, Average linkage (UPGMA), Centroid, Ward minimum variance (ward. D2), McQuitty, and Median in the dendextend package [22] in R for estimating the cophenetic correlation coefficient (CCC) between the distances and linkage methods pair. The CCC measures the correlation between the actual distances between the pairs of inbred lines and the cophenetic distance that represents the distance at which the inbreds are joined in the dendrogram. The CCC provides a measure of how well a hierarchical clustering tree represents the original distance matrix. With this technique, the higher the CCC the better is the pair of genetic distance and linkage method to identify appropriate heterotic pattern in our inbred using the Analysis of Phylogenetic and Evolution (APE) package [23]. The final dendrogram was constructed using the ggtree package [24] in R. Analysis of molecular variance (AMOVA) and genetic differentiation ($F_{ST}$) was done using GenAlEx 6.5 [25] to partition the variation between and within populations and pedigree of inbreds.

## Results

### Flowering property and genetic diversity indices for the 367 elite early white inbred population

The 367 inbred had considerable differences in their days to 50% flowering. The average days to 50% tasselling was 54 days after planting (range: 53—56 DAP) and 50% silking was 56 (range: 52—56). The diversity indices of the 376 inbred lines based on 1,904 SNP markers revealed that 57.6% have PIC greater than the average Polymorphism Information Content (PIC) of 0.39. The SNP markers on chromosome 9 has the highest PIC (0.41). The mean

heterozygosity expected/gene diversity (EH) averaged 0.37 (range: 0.07–0.50. Markers on chromosome 9 also had the highest level of observed heterozygosity (0.40), while markers on chromosome 1 had the lowest (0.35). Observed heterozygosity (OH) averaged 0.02 (range: 0.00 to 0.81). The MAF averaged 0.29 (range: 0.05–0.50). The MAF was highest in markers on chromosome 9 and lowest in markers on chromosomes 1 and 10. The Shannon (H) and Simpson (D) diversity indices averaged 6.86 (range: 6.81–7.03) and 949.09 (range: 899.31–1,085.95), respectively. The allelic richness averaged 787.70 (range: 728.29–1,322.49) (Table 1). The distribution of markers across the 10 chromosomes of the maize genome revealed that chromosome 10 had the least number of SNP markers (143), while chromosome 5 had the highest number (254) (Fig 1).

## Population structure of the 376 elite early white inbred lines assessed with 1904 SNP markers

From the STRUCTURE results, the value of LnP(D) increased continuously from K = 1 to 10; with an inflection point at 2 and 6 (Fig 2a and b). Based on the admixture model the 376 maize inbreds were grouped into two and six sub-populations, respectively (Fig 2c and d). The Introduction of an 80% membership probability threshold resulted in a 9.6% (36 inbreds) level of admixture at K = 2 and a 33% (125 inbreds) level of admixture at K = 6 (S2 and S3 Tables). Of the inbreds designated as admixture from K = 2 and K = 6, 11 inbreds were common to both K values. The admixture levels at K = 2 comprised 17 inbreds from TZEI 65 x ENT 11 and 19 inbreds from DTE STR-W Syn Pop C4. At K = 6 the admixture comprised 8 inbreds from TZEI 65 x ENT 11, 76 inbreds from DTE STR-W Syn Pop C4, and 41 inbreds from TZE-W Pop DT C5 STR C5. The visualization of the underlying population structure using PCA (Fig 3), elbow, Silhouette, and Gap statistics methods (S1 Fig) however, revealed three sub-populations. In addition, the pattern of the genetic stratification from PCA showed that sub-populations 1 had greater genetic closeness with sub-population 2 than with sub-population 3. Furthermore, sub-population 2 was stratified into two subgroups as observed in the PC1 and 2-plot view (Fig 3).

**Table 1. Diversity indices of the 376 elite early maturing white maize inbred lines based on 1,904 SNP markers.**

| Chromosome | OH | EH | MAF | PIC | H | 1/D | S |
|---|---|---|---|---|---|---|---|
| CHR1 | 0.02 | 0.35 | 0.27 | 0.38 | 4.61 | 100.19 | 83.39 |
| CHR2 | 0.02 | 0.37 | 0.3 | 0.38 | 4.55 | 91.19 | 79.95 |
| CHR3 | 0.02 | 0.37 | 0.28 | 0.39 | 4.74 | 114.2 | 94.43 |
| CHR4 | 0.03 | 0.38 | 0.29 | 0.4 | 4.62 | 101.53 | 85.04 |
| CHR5 | 0.02 | 0.37 | 0.28 | 0.39 | 4.83 | 125.08 | 102.79 |
| CHR6 | 0.03 | 0.37 | 0.3 | 0.38 | 4.42 | 82.58 | 69.18 |
| CHR7 | 0.02 | 0.38 | 0.3 | 0.39 | 4.51 | 86.37 | 71.28 |
| CHR8 | 0.02 | 0.38 | 0.28 | 0.39 | 4.57 | 96.6 | 79.77 |
| CHR9 | 0.03 | 0.4 | 0.31 | 0.41 | 4.38 | 79.58 | 67.28 |
| CHR10 | 0.02 | 0.37 | 0.27 | 0.39 | 4.28 | 71.89 | 59.72 |
| Min | 0 | 0.07 | 0.05 | 0 | 6.81 | 899.31 | 728.29 |
| Average | 0.02 | 0.37 | 0.29 | 0.39 | 6.86 | 949.09 | 787.70 |
| Max | 0.81 | 0.5 | 0.5 | 0.5 | 7.03 | 1,085.95 | 1,322.49 |

OH; observed heterozygosity, EH; expected heterozygosity, MAF; minor allele frequency, PIC; polymorphism information content, H; Shannon diversity index, 1/D; Inverse Simpson diversity index, S; Allele richness.

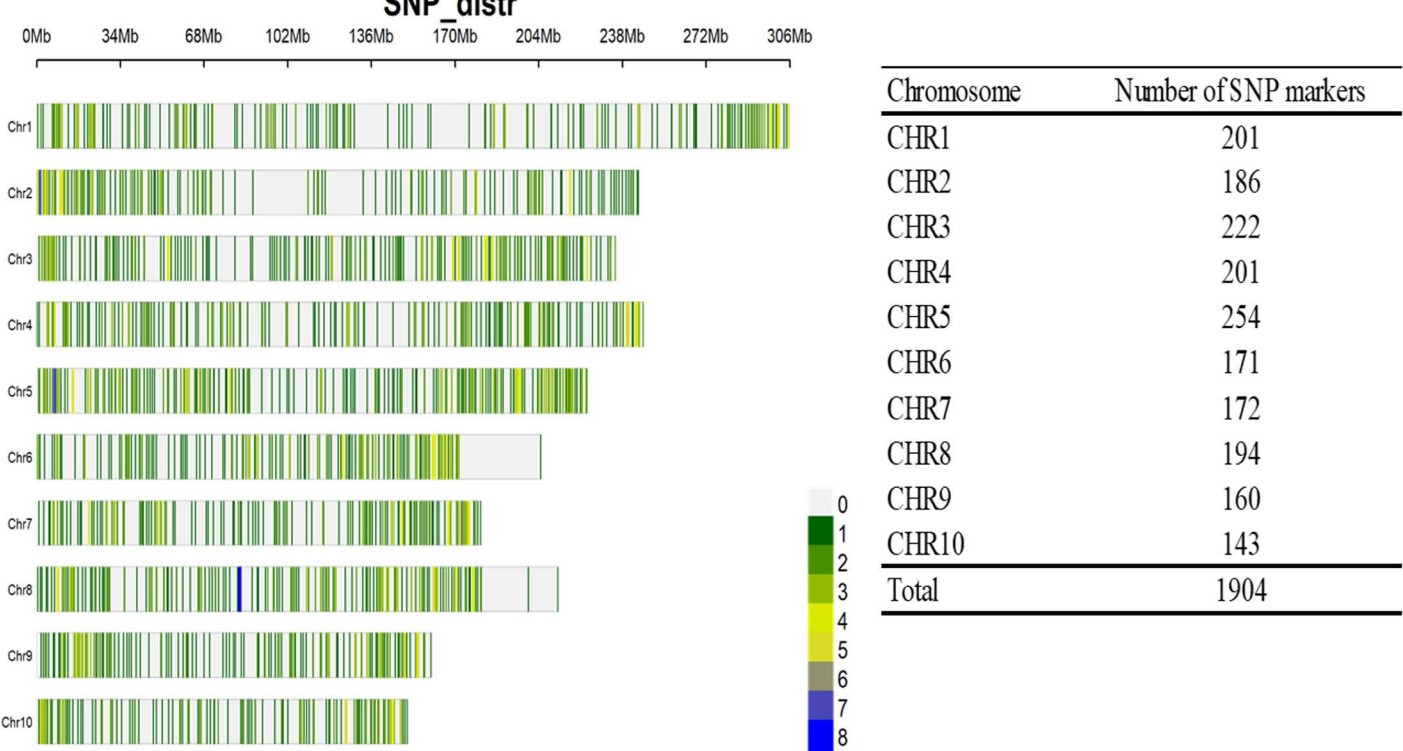

**Fig 1. Distribution of the 1904 SNP marker across the 10 chromosomes of the maize genome.** The number in the key indicated the quantity of SNP marker corresponding to the colour pattern depicted in the plot. Grey; no markers, dark to light green; 1–3 SNPs, light to dark yellow; 4–6 SNPs, light and deep blue; 7 and 8 SNPs, respectively.

## Defining the best clustering method and heterotic groups of 376 early white inbred lines

The cophenetic correlation coefficient (CCC) of the pair of genetic distance matrices and hierarchical clustering linkage methods presented in Table 2 showed that the IBS distance combined with the average linkage (UPGMA) method had the highest CCC (0.97). The clustering was therefore constructed with the UPGMA method using the IBS matrix. The IBS genetic dissimilarity/distance (GD) displayed a range of estimates from 0.002 between TZEI 2772 and TZEI 2761 to 0.372 between TZEI 2273 and TZEI 2832 with an average of 0.303. The HC assigned the 376-elite white inbreds to three major heterotic groups (Fig 4). One hundred and eighty elite inbreds were assigned to heterotic group 1 (red) with an average GD of 0.302. This group had the largest number of cluster members. In this heterotic group, the lowest GD was between the inbred pair TZEI 2772 and TZEI 2761 (0.002), while the maximum GD (0.371) was between the inbred pair TZEI 2832 and TZEI 2273. Heterotic group 2 (green) comprised 137 inbreds with an average GD of 0.302. The lowest GD for this heterotic group was between the inbred pair TZEI 2477 and TZEI 2490 (0.012), while the highest GD (0.369) was between the inbred pair TZEI 2273 and TZEI 2748. Heterotic group 3 (blue) had the least number of cluster members (59 inbreds) with an average GD of 0.301. The lowest GD (0.013) was between the inbred pair TZEI 2259 and TZEI 2548, while the highest GD (0.366) was between the inbred pair TZEI 2271 and TZEI 2729 (Table 3). Considering a 35% dissimilarity among inbreds from opposing heterotic groups as a benchmark, 661 hybrid combinations were

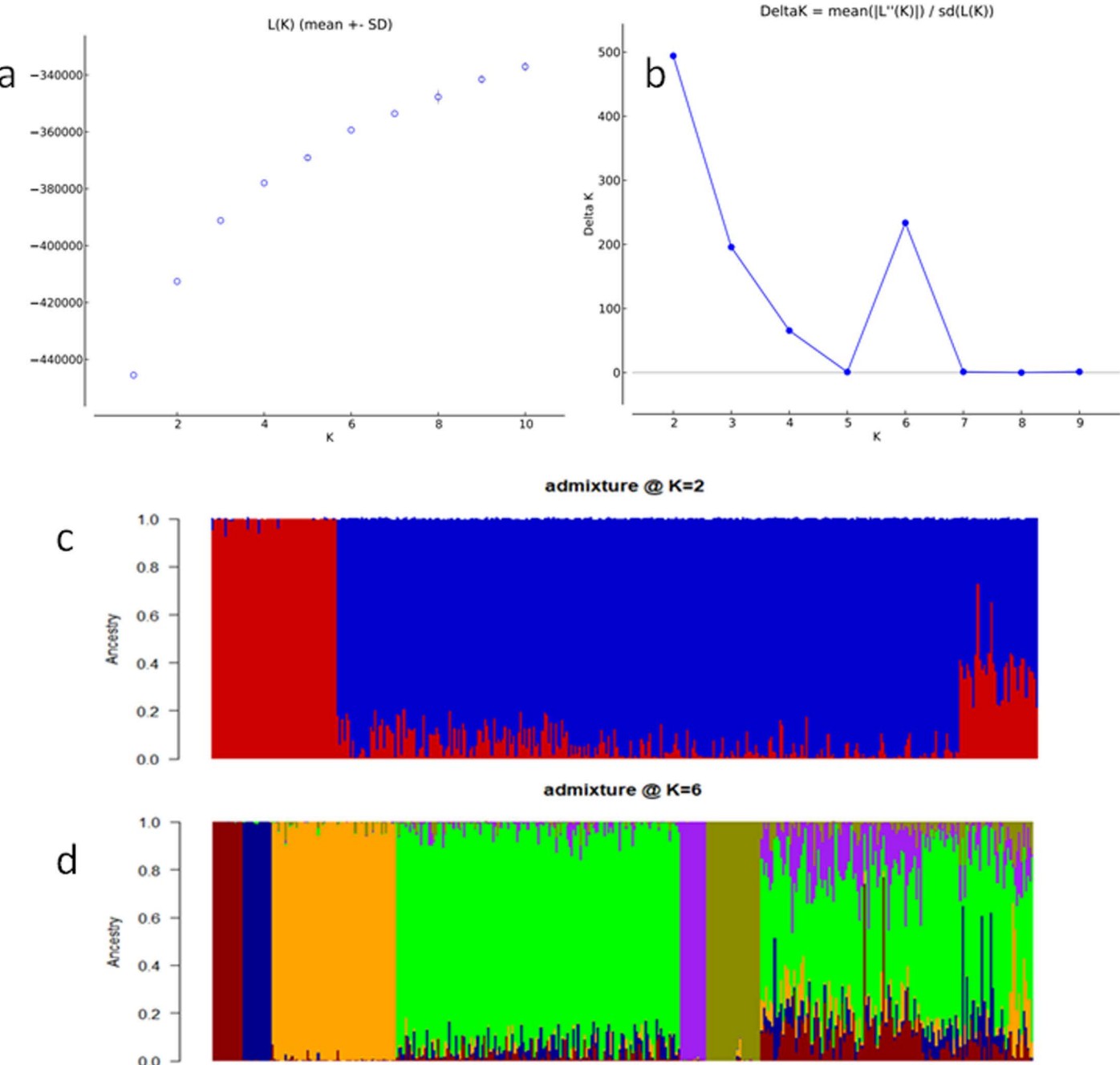

**Fig 2. Population structure of 376 elite maize inbreds using 1904 SNP markers revealing two inflections.** (a) Log probability of the Data, (b) Inflection showing optimum sub-populations K, (c) plot view of optimum sub-population @ K = 2 with red and blue representing sub-populations 1 and 2, respectively, and (d) plot view of optimum sub-population @ K = 6 with red, blue, orange, green, purple, and yellow representing sub-populations 1—6, respectively.

identified. These hybrids comprised 130 hybrid combinations between heterotic groups 1 and 2, 396 hybrid combinations between heterotic groups 1 and 3, and 135 hybrid combinations between heterotic groups 2 and 3 (Fig 5).

The PCA plot of the 376-elite inbreds based on pedigree information revealed the presence of common alleles among the source populations DTE STR-W Syn Pop C4, TZEI 65 x ENT

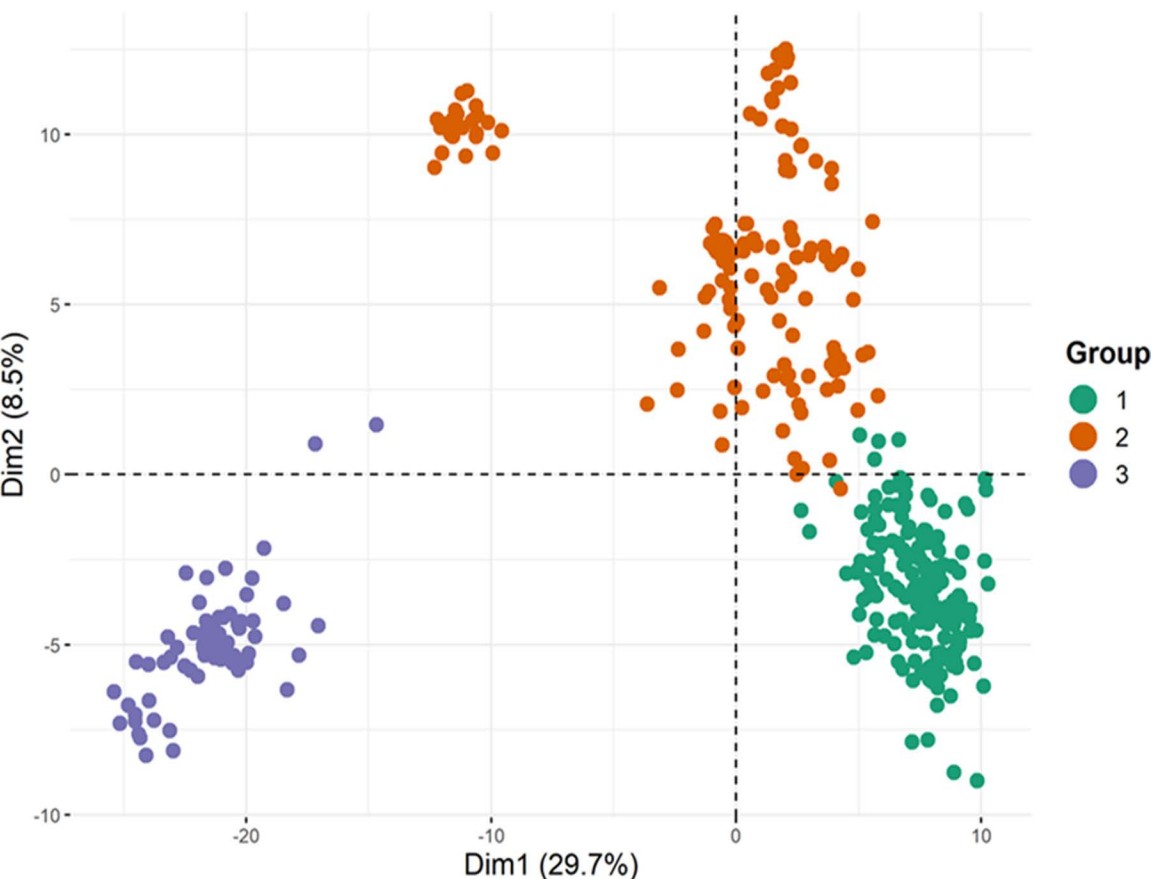

**Fig 3. Principal component analysis showing the three groups or subpopulations in the 376 maize inbreds.** Group one (green), group two (orange), and group three (purple).

**Table 2. Cophenetic correlation coefficients among the clustering linkage methods and genetic distance matrices employed to determine the best heterotic grouping methods for the 376 early-maturing white inbred lines.**

| Genetic distance Matrix | Hierarchical Clustering methods | | | | | | |
|---|---|---|---|---|---|---|---|
| | Average | Centroid | Complete | McQuitty | Median | Single | ward.D2 |
| Euclidean | 0.95 | 0.21 | 0.93 | 0.95 | 0.20 | 0.91 | 0.53 |
| IBS | **0.97** | 0.93 | 0.93 | 0.94 | 0.90 | 0.77 | 0.88 |
| Jaccard | 0.94 | 0.50 | 0.92 | 0.94 | 0.39 | 0.90 | 0.58 |
| Gower | 0.94 | 0.57 | 0.92 | 0.94 | 0.42 | 0.91 | 0.63 |

11, and TZE-W Pop DT C5 STR C5 (S1 Fig). Heterotic group 1 comprised 47.6% of all the inbred lines studied with 94.4% derived from population TZE- W Pop DT C5 STR C5 and 5.6% derived from population DTE STR-W Syn Pop C4. Heterotic group 2 constituted 36.7% of all the inbred lines studied and comprised inbred lines derived from all the three source populations. Inbred lines derived from populations DTE STR-W Syn Pop C4, TZE-W Pop DT C5 STR C5 and TZEI 65 × ENT 11 constituted 79.0%, 5.8% and 15.2% of heterotic group 2. Inbred lines derived from TZEI 65 × ENT 11 made up 66.1% of the inbred lines derived from heterotic group 3, while the remaining inbred s were derived from DTE STR-W Syn Pop C4

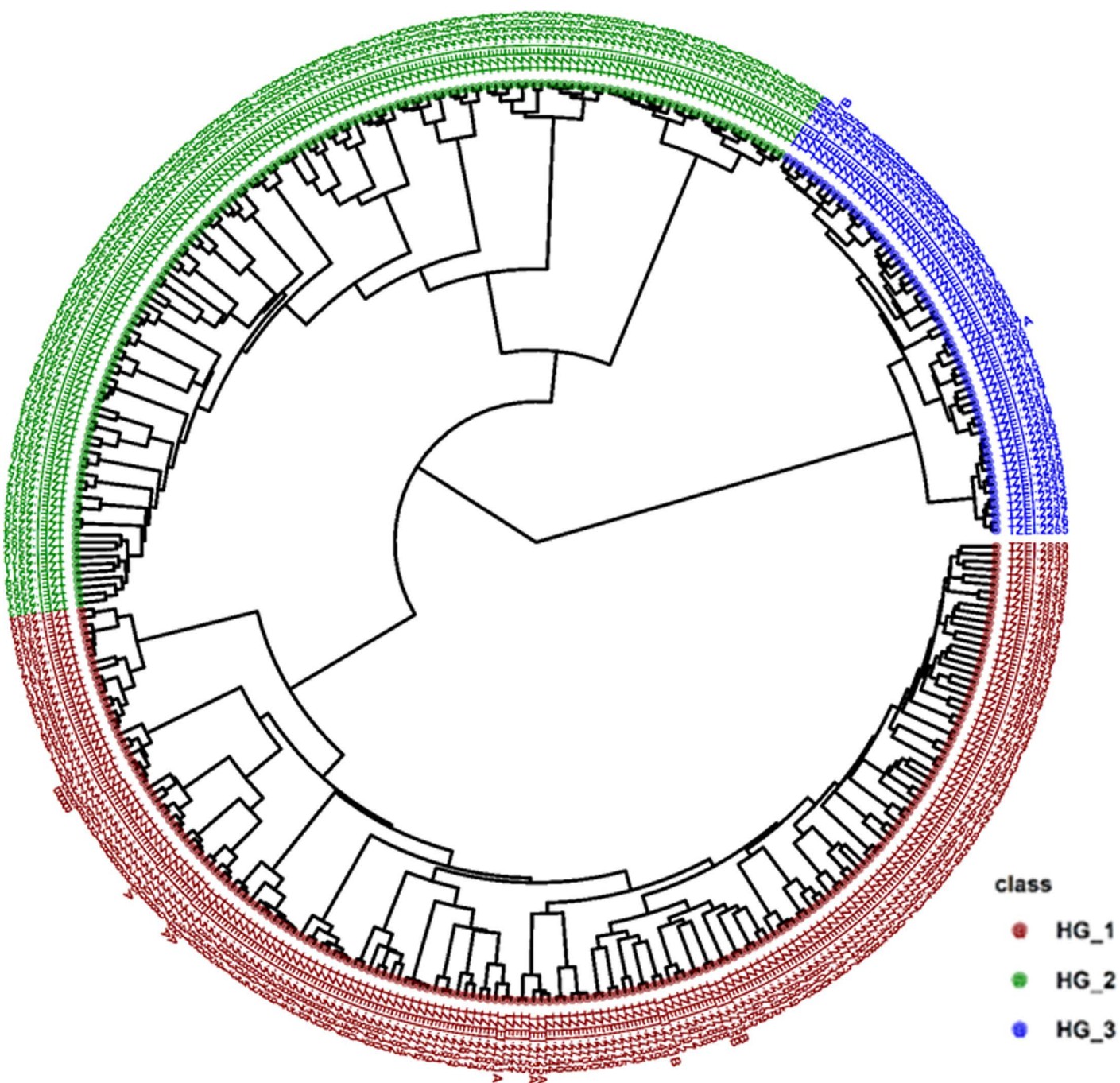

**Fig 4. Heterotic grouping of 376 early-maturing white maize inbreds showing three distinct groups.** HG_1 means heterotic group one, HG_2 means heterotic group two, and HG_3 means heterotic group three.

(33.9%). The three heterotic groups shared inbred lines derived from DTE STR-W Syn Pop C4. The pedigree-based phylogenetic tree assigned 172 of 180 inbreds extracted from TZE-Pop DT C5 STR C5 into heterotic group 1, 109 of 138 inbreds extracted from DTE STR-W Syn Pop C4 into heterotic group 2, and 39 of 59 inbreds extracted from TZEI 65 x ENT 11 into heterotic group 3 (Figs 6 and S2).

**Table 3. Genetics distance from 1904 DArT-derived SNP markers characterizing the three heterotic groups defined for the 376 early maturing white inbred lines.**

| | HG_1 | HG_2 | HG_3 |
|---|---|---|---|
| Minimum | 0.002 (TZEI 2772 × TZEI 2761) | 0.012 (TZEI 2477 × TZEI 2490) | 0.013 (TZEI 2259 × TZEI 2548) |
| Maximum | 0.371 (TZEI 2832 × TZEI 2273) | 0.369 (TZEI 2273 × TZEI 2748) | 0.366 (TZEI 2271 × TZEI 2729) |
| Average | 0.302 | 0.302 | 0.301 |

HG_1, heterotic group one; HG_2, heterotic group two; HG_3, heterotic group three

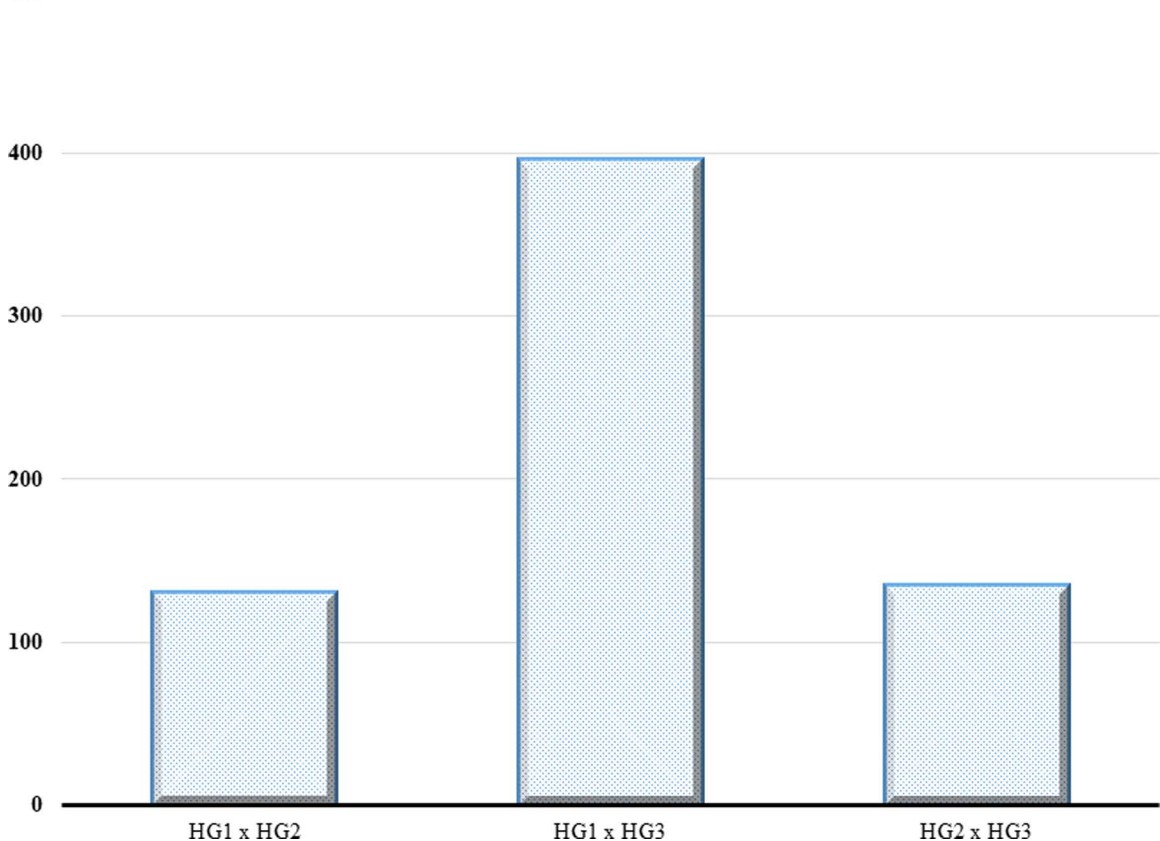

**Fig 5. Potential hybrid combinations from opposing heterotic groups identified based on genetic dissimilarity of 35%.** HG_1, heterotic group one; HG_2, heterotic group two; and HG_3, heterotic group three.

## Analysis of molecular variance and genetic differentiation

The AMOVA results (Table 4) based on the heterotic group and pedigree information captured the spread of genetic variation between and within the heterotic groups and pedigrees. Based on the heterotic structure, the results revealed a genetic variation of 10% among heterotic groups and 90% within heterotic groups, with a moderate level of genetic differentiation (0.099). Among pedigrees, genetic variation was 7%, while within-pedigrees variation was 93%, with a moderate level of genetic differentiation (0.073).

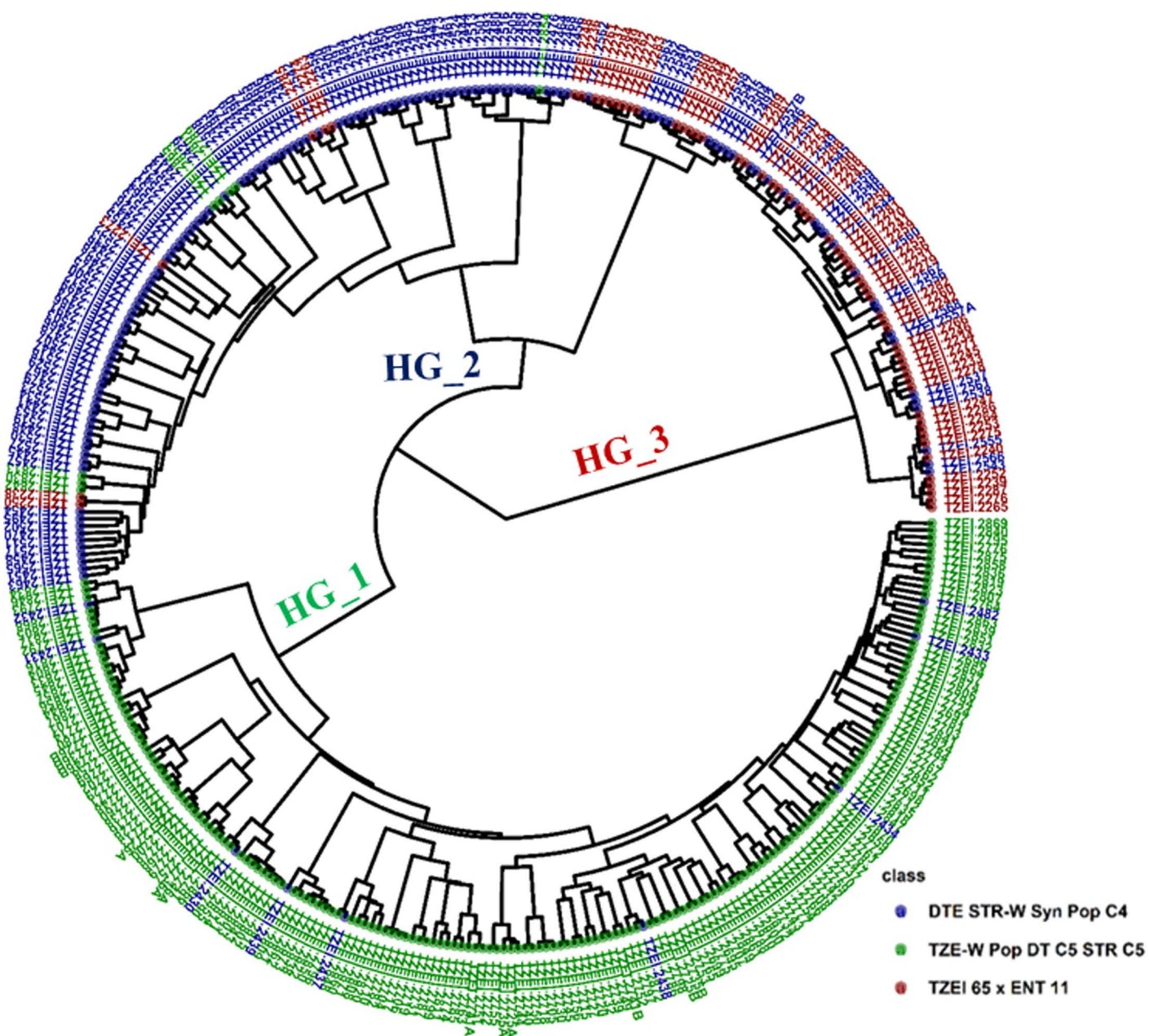

**Fig 6. Phylogenetic relatedness among 376 early-maturing white maize inbreds based on the source populations.** HG_1 means heterotic group one, HG_2 means heterotic group two, while HG_3 means heterotic group three.

Heterotic groups 1 and 2 had the lowest level of differentiation (0.059) and the highest gene flow (7.999), while the differentiation was highest (0.183) between heterotic groups 3 and 1 with the lowest gene flow (2.225). Based on the source populations, DTE STR-W Syn Pop C4 and TZEI 65 x ENT 11 had the least pairwise differentiation (0.011) with the highest level of gene flow (44.468), while DTE STR-W Syn Pop C4 and TZE-W Pop DT C5 STR C5 had the highest differentiation (0.090) with the lowest gene flow (5.026) (Table 5).

**Table 4. Analysis of molecular variance and genetic differentiation among 376 early-maturing maize inbred lines assessed using 1904 SNP markers.**

| Source | DF | SS | MS | Est. Var. | % | Fst | P |
|---|---|---|---|---|---|---|---|
| Among HG | 2 | 9,556.319 | 4,778.159 | 38.403 | 10% | 0.099 | 0.001 |
| Within HG | 373 | 130,269.094 | 349.247 | 349.247 | 90% | | |
| Total | 375 | 139,825.412 | | 387.650 | 100% | | |
| Source | DF | SS | MS | Est. Var. | % | Fst | P |
| Among Pedigree | 2 | 7,218.986 | 3,609.493 | 28.188 | 7% | 0.073 | 0.001 |
| Within Pedigree | 373 | 132,606.426 | 355.513 | 355.513 | 93% | | |
| Total | 375 | 139,825.412 | | 383.701 | 100% | | |

DF: degrees of freedom; HG; Heterotic group; SS: sum of squares; MS: mean squares; Est.Var: estimated variance; %: percentage variance; Fst: Pairwise differentiation; P: significance level

**Table 5. Pairwise population differentiation based on population structure and pedigree information of the 376-elite white inbreds.**

| Source | Group 1 | Group 2 | Group 3 |
|---|---|---|---|
| Heterotic group 1 | 0.000 | 0.001 | 0.001 |
| Heterotic group 2 | 0.059 | 0.000 | 0.001 |
| Heterotic group 3 | 0.183 | 0.098 | 0.000 |
| | DTE STR-W Syn Pop C4 | TZEI 65 x ENT 11 | TZE-W Pop DT C5 STR C5 |
| DTE STR-W Syn Pop C4 | 0.000 | 0.001 | 0.001 |
| TZEI 65 x ENT 11 | 0.011 | 0.000 | 0.001 |
| TZE-W Pop DT C5 STR C5 | 0.090 | 0.077 | 0.000 |

## Discussion

### Genetic diversity indices

Genetic diversity forms an important underlying structure for genetic enhancement in any breeding program. In this study, 1904 SNP markers were used to assess the level of genetic diversity, population structure and identify the best genetic distance by linkage methods for assigning the 367 inbreds in heterotic groups. The mean marker Polymorphism Information Content (PIC) showed that the markers were appropriate for assessing the genetic diversity of the early elite inbreds. Polymorphism information content explains the usefulness of a genetic marker in linkage and diversity studies as it reveals the ability of markers to differentiate and discriminate among genotypes. The average marker PIC obtained in this study was comparable to those reported in tropical maize germplasm by some previous studies [26–28] and higher than those reported in some other studies [10,29,30] The average marker minor allele frequency (MAF) suggested the presence of rare alleles in substantial proportions in the maize inbred population. Adu et al. [26] reported similar findings. The expected heterozygosity indicated the probability of individuals possessing different alleles at a given SNP locus. The high estimates of expected heterozygosity as observed in this study suggested high genetic diversity within the population. The average Shannon and inverse Simpson indices recorded suggested the existence of a wide range of genetic variation, which is essential for IITA maize breeding programs for hybrid development. The alpha index, a synonym of species richness that measures the number of different genotypes present in the population suggests a large number of unique genotypes within the 367 inbred lines. Several authors have used these diversity parameters to assess the level of genetic diversity for breeding optimization [13,31].

## Population structure of 376 elite early white inbred population using SNP markers

Based on the 1904 SNP markers used to investigate the population structure and the relationship patterns of the elite white inbred population, a discord was observed from the outputs of the Bayesian statistics approach in STRUCTURE software and the principal component analysis (PCA). While STRUCTURE concluded that, the optimum number of sub-populations was two and six, the alternative approach concluded that the optimum number of sub-populations was three. Additionally, PCA revealed closer genetic proximity between subpopulations 1 and 2, suggesting they could have been considered a single subpopulation by the Bayesian statistics approach in STRUCTURE software. The percentage of admixture resulting from K = 2 (sub-populations) was far lower than compared to when k = 6 suggesting that admixture levels increase with an increase in the level of stratification. The level of admixture in this study at both K values defined as optimum did not translate to genetic contamination but rather inbred lines that combined genomic background or information  of more than one sub-populations  [10]. In the present study, three sub-populations were considered optimum. Knowledge of population stratification is essential for explaining differences in the genetic architecture of any population though spatial and gene exchange isolation could affect the output [32].

## Heterotic pattern of 376 elite early inbred population using SNP markers

The distance matrices and clustering methods used for heterotic classification have implications on the results owing to the strengths and weaknesses attributable to different matrices and methods [13]. Our study attempted to identify the best clustering method and distance matrix for classifying the 367 inbred lines into heterotic groups. We use the concept of the cophenetic correlation coefficient calculated as the correlation between the original distance matrix (actual distance between two inbreds) and the cophenetic distance matrix (distance at which two inbreds are joined together in the dendrogram). We observed that the use of the average linkage method for the IBS distance matrix provided the best clustering quality solution for classifying our 376 inbreds in heterotic groups. Darkwa et al. [13] and Bagheri et al. [33] reported that by evaluating the cophenetic correlation, breeders can assess the quality of the clustering and choose the most appropriate clustering method for classification of breeding germplasm into genetic groups. In hybrid maize breeding, using appropriate distance and linkage method can improve the precision of heterosis prediction as it ensures that parents with sufficient genetic divergence are identified for potential crossings. Additionally, by accurately assessing genetic relatedness, breeders can prioritize crosses with the highest potential for heterosis and thereby optimizing resource allocation [27].

The heterotic grouping of the elite inbreds was largely consistent with the pedigree and selection history of the inbreds. It is noteworthy that each of the heterotic groups had some inbreds from more than one source population. This observation suggested that the source populations from where the 376 elite inbreds were developed could have shared alleles, which is a common observation in breeding programs. The number of heterotic groups observed in this study (three) was similar to that reported by other Adu et al. [26] for a population of 94 white and yellow elite early inbreds using DArTseq SNP markers. In another study conducted by Badu-Apraku et al. [27] with a panel of 439 early and extra-early maize inbreds derived from diverse source populations, four heterotic groups were identified. The authors reported that the ideal heterotic groups for each maturity group and kernel colour should be two, labelled A and B or at most three labelled A, B, and C (the mixed group).

## Potential hybrid combinations and genetic differentiation

The paradigm shift from open-pollinated varieties to hybrid maize production in the West African sub-region since the 2000s has emphasized the development of inbred lines and their classification into heterotic groups for hybrid maize production [27]. Several methods have been employed at IITA to classify early maize inbred lines, including phenotypic data, specific combining ability, combining ability of multiple traits, and more recently, DNA-based markers. Heterotic groups are continuously refined as new inbred lines are developed from heterotic and broad-based populations [8,34]. From the three heterotic groups, we identified 661 potential hybrid combinations that can produce outstanding hybrids (Fig 5). Of this large number the best 10 were TZEI 2273 × TZEI 2832, TZEI 2273 × TZEI 2748, TZEI 2271 × TZEI 2729, TZEI × TZEI 2725A, TZEI 2276 × TZEI 2748, TZEI 2273 × TZEI 2799, TZEI 2273 ×TZEI 2794, TZEI 2273 × TZEI 2566, TZEI 2276 × TZEI 2760, and TZEI 2234 × TZEI 2701. These hybrids have genetic distance ranging from 0.35 to 0.37. Hybrid maize production exploits heterosis, the enhanced vigour observed in progeny from two complementary inbred lines. Inbreds from different heterotic groups exhibit higher levels of heterosis compared to those from the same group. The latter can only exhibit high heterosis when crossed in a three-way hybrid combination with inbreds differing in specific characteristics [27].

The analysis of molecular variance revealed higher levels of genetic variability within the heterotic groups than among the heterotic groups with a moderate estimate of genetic differentiation. A similar pattern of observation exists based on the pedigree or source populations. A deeper dive into the pairwise genetic differentiation among the observed heterotic groups revealed that heterotic groups I and III were characterized by high genetic differentiation and as such were suitable for the exploration of heterosis for early-maturity white hybrid maize production. These heterotic groups comprised inbreds from the source populations TZEI 65 x ENT 11 and TZE-W Pop DT C5 STR C5. The justification for this observation could be the involvement of ENT 11 as a parental inbred of the heterotic population. ENT 11 is a CIMMYT inbred with a different genetic background compared to TZEI 65 or the broad-based populations developed by IITA. Inbreds derived from TZEI 65 x ENT 11 presumably acquired unique haplotype blocks through recombination during inbreeding, leading to greater genetic distance from those developed solely from IITA germplasm that mostly originate from the US corn belts [27,34]. Similar, though less pronounced, genetic differentiation was observed between Groups II and III. Conversely, Groups I and II exhibited the lowest $F_{ST}$ values, suggesting they may not display significant heterosis in hybrid combinations. Inbreds within these groups originated from the IITA broad-based populations TZE-W Pop DT C5 STR C5 and DTE STR-W Syn Pop C4. Based on the understanding obtained from several reports, genetic differentiation values between 0 and 0.05 suggested low differentiation, 0.06 and 0.15 as moderate, 0.16 and 0.25 as high, and above 0.25 as very high [35] Understanding pairwise genetic differentiation is crucial for breeding cross-pollinated crops like maize, as it facilitates efficient exploitation of hybrid vigor for enhanced selection gains [36].

The observed gene flow pattern mirrored the inverse trend of genetic differentiation. The lowest gene flow, indicative of reduced allele sharing, occurred between groups, the most genetically differentiated groups, making them suitable for hybrid production. Conversely, the highest gene flow, suggesting substantial shared alleles, was observed between group I and II, potentially leading to lower heterosis in hybrid combination. Gene flow exceeding unity signifies sufficient gene flow [34], explaining why the admixture-based model implemented in STRUCTURE grouped them into a single sub-population.

This study provides valuable data to guide early maturity white hybrid maize development at the IITA maize improvement program. Future efforts should focus on aligning these findings with existing heterotic groups and implementing strategies to refine these groups

for optimal genetic gain and cost-effective hybrid production. Notably, the impracticality of managing six distinct heterotic groups for early maturity white maize, as suggested by the K = 6 outcome in STRUCTURE, necessitates a streamlining approach for efficient breeding practices.

## Conclusions

The study effectively employed 1,904 SNP markers to characterize genetic diversity among 376 elite early maturing white maize inbred lines. The marker diversity indices, Shannon, Simpson, and Alpha indices all showed the presence of high genetic diversity among the 367 elite early-white inbred population studied. The IBS distance matrix and average linkage clustering methods provided the best clustering quality used to define the heterotic pattern of the 367 elite early-white inbred population. Three distinct heterotic groups, each comprising inbreds from more than one source population characterized the 367 inbreds. Of the three heterotic groups, inbreds from groups I and III showed the greatest heterotic potential. The information on the diversity and heterotic pattern observed in this study offers a promising foundation for developing superior maize hybrids. By capitalizing on the heterotic potentials, breeders in the IITA-MIP can prioritize crosses with the greatest potential for hybrid vigor, thereby accelerating the development of high-yielding, stress-tolerant hybrid maize genotypes. To maximize the impact of these findings, policymakers should invest in research to further refine heterotic group identification methods and support the development of marker-assisted selection tools.

## Supporting information

**S1 File. Supplementary S1–S4 Tables.**
(DOCX)

**S1 Fig. Optimal sub-population based on Elbow method (upper), Silhouette method (middle), and Gap statistics method (lower).**
(TIF)

**S2 Fig. Pedigree-based PCA scatter plot of the 376 early maturing maize inbred lines along PC1 and PC2 using 1904 SNP markers.**
(TIF)

## Acknowledgments

The authors are grateful to the African Union Commission and the African Development Bank through the Pan African University Life and Earth Science Institute (PAULESI) for providing academic scholarship to the first author that led to these findings. The entire staff of the University of Department of Crop and Horticultural Sciences, University of Ibadan for their tutelage and the research staff of the Maize Improvement Program of the International Institute of Tropical Agriculture, Ibadan, Nigeria, for their unwavering support in germplasm provision and leaf sampling.

## Author contributions

**Conceptualization:** Hellen Mawia Mukiti, Baffour Badu-Apraku, Idris Ishola Adejumobi, John Derera, Ayodeji Abe.

**Data curation:** Idris Ishola Adejumobi.

**Formal analysis:** Idris Ishola Adejumobi.

**Funding acquisition:** Baffour Badu-Apraku, John Derera.

**Methodology:** Hellen Mawia Mukiti, Idris Ishola Adejumobi.

**Project administration:** Baffour Badu-Apraku, John Derera.

**Resources:** John Derera.

**Supervision:** Baffour Badu-Apraku, Idris Ishola Adejumobi, John Derera, Ayodeji Abe.

**Writing – original draft:** Hellen Mawia Mukiti.

**Writing – review & editing:** Hellen Mawia Mukiti, Baffour Badu-Apraku, Idris Ishola Adejumobi, John Derera, Ayodeji Abe.

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
