## [Decision Letter · Decision Letter 0]

6 Nov 2024

PONE-D-24-43431Assessment of genetic diversity, population structure, and clustering methods for heterotic grouping in early-maturing white maize (Zea mays L.) inbred linesPLOS ONE

Dear Dr. Adejumobi,

Thank you for submitting your manuscript to PLOS ONE. After careful consideration, we feel that it has merit but does not fully meet PLOS ONE’s publication criteria as it currently stands. Therefore, we invite you to submit a revised version of the manuscript that addresses the points raised during the review process. Please submit your revised manuscript by Dec 21 2024 11:59PM. If you will need more time than this to complete your revisions, please reply to this message or contact the journal office at plosone@plos.org . Please include the following items when submitting your revised manuscript:

We look forward to receiving your revised manuscript.

Kind regards,

Muhammad Abdul Rehman Rashid, PhD

Academic Editor

PLOS ONE

Journal Requirements:

“Bill and Melinda Gates Foundation through the Accelerated Genetic Gains and Stress Tolerant Maize for Africa Projects [OPP1134248].”

“The authors are grateful to the staff of the Maize Improvement Unit of the International Institute of Tropical Agriculture, Ibadan, Nigeria, the African Union Commission, and the African Development Bank through the Pan African University of Life and Earth Science Institute (PAULESI) for providing funds for the research. The authors are also grateful for the funding support of the Bill & Melinda Gates Foundation under the Accelerated Genetic Gains in Maize and Wheat (AGG) Projects.”

“Bill and Melinda Gates Foundation through the Accelerated Genetic Gains and Stress Tolerant Maize for Africa Projects [OPP1134248].”

“I have read the journal's policy, and the authors of this manuscript do not have any competing interests.”

5. Thank you for uploading your study's underlying data set. Unfortunately, the repository you have noted in your Data Availability statement does not qualify as an acceptable data repository according to PLOS's standards.

6. We notice that your supplementary files are included in the manuscript file. Please remove them and upload them with the file type 'Supporting Information'. Please ensure that each Supporting Information file has a legend listed in the manuscript after the references list.

Reviewers' comments:

Reviewer's Responses to Questions

**Comments to the Author**

1. Is the manuscript technically sound, and do the data support the conclusions?

Reviewer #1: Partly

Reviewer #2: Yes

Reviewer #3: Yes

Reviewer #4: Yes

2. Has the statistical analysis been performed appropriately and rigorously? 

Reviewer #1: Yes

Reviewer #2: Yes

Reviewer #3: Yes

Reviewer #4: Yes

3. Have the authors made all data underlying the findings in their manuscript fully available?

Reviewer #1: Yes

Reviewer #2: Yes

Reviewer #3: Yes

Reviewer #4: Yes

4. Is the manuscript presented in an intelligible fashion and written in standard English?

Reviewer #1: No

Reviewer #2: Yes

Reviewer #3: Yes

Reviewer #4: Yes

5. Review Comments to the Author

Reviewer #1: - The English needs native revision.

- The title needs to revised.

- The abstract is not precise to cover the research properly.

- The introduction needs revision with the newly related papers.

- In the material and methods needs more detail.

- The figure are not to the extent for publishing.

- The discussion has no proper critical challenging with the related references.

You may get the advantage of the following references throughout the text.

- Qalavand, F., et al 2022. Enzyme activity and population genetic structure analysis in wheat associated with resistance to Bipolaris sorokiniana-common root rot diseases. Phytochem. 200, 113208.

- Moatamedi, M., et al. 2018. Genetic variation of bread wheat accessions in response to the cereal cyst nematode, Heterodera filipjevi. Nematology, 20(9), 859-875.

- Sadeghpoor, N., (2023) Assessing genetic diversity and population structure of Iranian melons (Cucumis melo) collection using primer pair markers in association with resistance to Fusarium wilt. Functional Plant Biology 50, 347-362.

- Moghaddam, G.A, Nasr Esfahani, M., Rezayatmand, Z., Khozaei, M., 2022. Genomic markers analysis associated with resistance to Alternaria alternata (fr.) keissler—tomato pathotype, Solanum lycopersicum L. Breeding Science Preview. doi: 10.1270/jsbbs.22003.

- Bagheri L.M., et al. 2022. Screening for resistance and genetic population structure associated with Phytophthora capsici-pepper root and crown rot. Physiol. Mol. Plant Pathol.. 119, 101835. 10.1016/j.pmpp.2022.101924

Reviewer #2: The authors assessed genetic diversity among 376 white maize inbreds possessing multiple stress tolerance, and have carried our heterotic grouping for its effective use in the hybrid development program in maize. The study is well undertaken, and the manuscript is very well written. The manuscript can be accepted for publication except for a few MINOR REVISIONS below:

(1) Correct as 'Polymorphism Information Content' for PIC, throughout the manuscript.

(2) The choice of material is perfect as they are Sy inbreds possessing multiple stress tolerance. In section 2.1, mention whether any hybrids/ experimental hybrid combinations have already been identified from this panel.

(3) Present the 'Discussion' with suitable sub-headings for clear and easy understanding to the readers.

(4) In the Discussion, please add a paragraph on the 'promising hybrid combinations identified from the study' based on genetic distance, agronomic behavior, and stress tolerance.

(5) Since all the 376 inbreds belong to the early maturity group, present in the results the mean and range for days to 50% tasselling.

Reviewer #3: Reference for the method used for genotyping in section 2.3 should be included. In a number of cases references cited in the text have the bracket at the wrong point e.g. on page 16, third paragraph, line 15 ------reported by (Adu et al., 2019) the correct way is --------reported by Adu et al. (2019). The authors should check and correct wherever this error occurs.

Reviewer #4: Reviewers Report

The study reported in this manuscript attempted to study genetic variability and population structure in 378 elite inbred lines using 1904 SNP markers. The idea underlining the study was well conceptualized, the set objectives were measurable and achievable and the execution was appropriately carried out using adequate methodology. The results were well presented using good tables and figures.

Major shortcomings

1. The justification of this study is too weak. What warrants comparing different genetic distance methods and clustering methods. If hierarchical clustering method is the most popular, what is responsible for that? What are the disadvantages of the other methods. I expect the authors to dig deeper into the nitty gritty of the methods to identify the knowledge gap more clearly.

3. The comparison of the methods to identify the best method was not well presented. The authors claimed CCC was the basis but this is just a correlation coefficient, not a statistic to determine efficiency per se. More so, the CCC does not carry any test of significance in order to make inference.

Conclusion appears too wordy. It should be recast to capture the major findings based on the stated objectives.

3. All references cited in the text were are listed in the reference section except Badu-Apraku et al., 2021

Recommendation

Accepted subject to MINOR REVISION

6. PLOS authors have the option to publish the peer review history of their article (what does this mean? ). If published, this will include your full peer review and any attached files.

**Do you want your identity to be public for this peer review?** For information about this choice, including consent withdrawal, please see our Privacy Policy .

Reviewer #1: No

Reviewer #2: **Yes: ** Dr. Vignesh Muthusamy

Reviewer #3: **Yes: ** Prof. Pangirayi Tongoona

Reviewer #4: **Yes: ** Richard Olutayo AKINWALE

---

## [Author Response · Author response to Decision Letter 1]

11 Dec 2024

Optimizing Breeding Strategies for Early-Maturing White Maize through Genetic Diversity and Population Structure

Hellen Mawia Mukiti1, 2, Baffour Badu-Apraku1, Ayodeji Abe3,

Idris Ishola Adejumobi1*, John Derera1

Response

We have ensured the manuscript meets PLOS ONE’s style requirements including those for file naming.

“Bill and Melinda Gates Foundation through the Accelerated Genetic Gains and Stress Tolerant Maize for Africa Projects [OPP1134248].”

Response

The funder Bill and Melinda Gates Foundation will cover the manuscript’s article processing fees (decision to publish) and we have included this role in the cover letter submitted alongside the revised manuscript.

“The authors are grateful to the staff of the Maize Improvement Unit of the International Institute of Tropical Agriculture, Ibadan, Nigeria, the African Union Commission, and the African Development Bank through the Pan African University of Life and Earth Science Institute (PAULESI) for providing funds for the research. The authors are also grateful for the funding support of the Bill & Melinda Gates Foundation under the Accelerated Genetic Gains in Maize and Wheat (AGG) Projects.”

“Bill and Melinda Gates Foundation through the Accelerated Genetic Gains and Stress Tolerant Maize for Africa Projects [OPP1134248].”

Response

Thank you. We have removed the funding-related information in the acknowledgment and also recast the financial disclosure in the revised version of the manuscript. We have also included this modification in the cover letter.

“I have read the journal's policy, and the authors of this manuscript do not have any competing interests.”

Response

Thank you. We will update the competing interest with the statement “The authors have declared that no competing interests exist” while submitting the revised manuscript and include the same in the cover letter.

5. Thank you for uploading your study's underlying data set. Unfortunately, the repository you have noted in your Data Availability statement does not qualify as an acceptable data repository according to PLOS's standards.

Response

Thank you for the suggestion. We have put the dataset in a figshare account with the URL (https://figshare.com/account/items/27968589/edit). It is also retrievable as https://docs.google.com/document/d/18BnsKpVquOfnCCS0x41vk3aCKnof7cuj6OHyBlOuYEI/edit?tab=t.0.

6. We notice that your supplementary files are included in the manuscript file. Please remove them and upload them with the file type 'Supporting Information'. Please ensure that each Supporting Information file has a legend listed in the manuscript after the references list.

Response

We have ensured that each supporting document has a legend in the manuscript.

Reviewer #1: - The English needs native revision.

(1) The title needs to revised.

Response

Thank you for the comment. With the approval of all authors, we have refined the topic to be more impactful as requested in the revised version of the manuscript.

(2) The abstract is not precise to cover the research properly.

Response

Thank you for the comment. We have modified the abstract to be more precise and focus more on the research in the revised version of the manuscript as requested.

(3) The introduction needs revision with the newly related papers.

Response

Thank you for the comment. We have added more recent references to the introduction and also revised the section to improve readability and chronology to the readers in the modified version of the manuscript as requested.

(4) In the material and methods needs more detail.

Response

We appreciate the comment. We have added more information to the materials and method in the modified version of the manuscript as requested.

(5) The figure are not to the extent for publishing.

Response

We appreciate the comment. We have replaced the manuscript figures with those of higher resolution to ensure they meet the journal requirement as requested.

(6) The discussion has no proper critical challenging with the related references.

Response

We appreciate the comments and the suggested articles to improve the quality of our manuscript. We have reviewed the discussion of the manuscript to address the critical challenges highlighted in our manuscript as requested.

You may get the advantage of the following references throughout the text.

- Qalavand, F., et al 2022. Enzyme activity and population genetic structure analysis in wheat associated with resistance to Bipolaris sorokiniana-common root rot diseases. Phytochem. 200, 113208.

- Moatamedi, M., et al. 2018. Genetic variation of bread wheat accessions in response to the cereal cyst nematode, Heterodera filipjevi. Nematology, 20(9), 859-875.

- Sadeghpoor, N., (2023) Assessing genetic diversity and population structure of Iranian melons (Cucumis melo) collection using primer pair markers in association with resistance to Fusarium wilt. Functional Plant Biology 50, 347-362.

- Moghaddam, G.A, Nasr Esfahani, M., Rezayatmand, Z., Khozaei, M., 2022. Genomic markers analysis associated with resistance to Alternaria alternata (fr.) keissler—tomato pathotype, Solanum lycopersicum L. Breeding Science Preview. doi: 10.1270/jsbbs.22003.

- Bagheri L.M., et al. 2022. Screening for resistance and genetic population structure associated with Phytophthora capsici-pepper root and crown rot. Physiol. Mol. Plant Pathol.. 119, 101835. 10.1016/j.pmpp.2022.101924

Reviewer #2: The authors assessed genetic diversity among 376 white maize inbreds possessing multiple stress tolerance, and have carried our heterotic grouping for its effective use in the hybrid development program in maize. The study is well undertaken, and the manuscript is very well written. The manuscript can be accepted for publication except for a few MINOR REVISIONS below:

(1) Correct as 'Polymorphism Information Content' for PIC, throughout the manuscript.

Response

The correction has been effected as 'Polymorphism Information Content' for PIC as requested in the revised manuscript.

(2) The choice of material is perfect as they are S7 inbreds possessing multiple stress tolerance. In section 2.1, mention whether any hybrids/ experimental hybrid combinations have already been identified from this panel.

Response

We appreciate your comment. From our findings, we have identified 661 potential hybrid combinations. Of this number, we have completed the development of over 300 testcross hybrids from the inbreds in the 661 combinations planned for performance evaluation under stress (drought and artificial Striga infestation) and non-stress conditions in 2025. The crosses made at S4 were to guide us in identifying lines with good GCA effects for selection and advancement to a fixed state through inbreeding.

(3) Present the 'Discussion' with suitable sub-headings for clear and easy understanding to the readers.

Response

We appreciate your comment. We have included suitable sub-headings for clear and easy understanding in the revised manuscript as requested.

(4) In the Discussion, please add a paragraph on the 'promising hybrid combinations identified from the study' based on genetic distance, agronomic behavior, and stress tolerance.

Response

We appreciate your comment. We have included some of the most promising hybrids combinations from the 661 potential hybrid combinations identified in the study in the revised manuscript as requested.

(5) Since all the 376 inbreds belong to the early maturity group, present in the results the mean and range for days to 50% tasselling.

Response

We appreciate your comment. We have included information relating to the mean and range for days to taselling and silking in the results section of the revised manuscript as requested.

Reviewer #3:

(1) Reference for the method used for genotyping in section 2.3 should be included.

Response

We appreciate your comment. We have included the reference for the genotyping method used for the study in the revised manuscript as requested.

(2) In a number of cases references cited in the text have the bracket at the wrong point e.g. on page 16, third paragraph, line 15 ------reported by (Adu et al., 2019) the correct way is --------reported by Adu et al. (2019). The authors should check and correct wherever this error occurs.

Response

We appreciate your comment. We have made the necessary corrections to the in-text citation style to meet the requirements of the journal in the revised manuscript.

Reviewer #4: Reviewers Report

The study reported in this manuscript attempted to study genetic variability and population structure in 378 elite inbred lines using 1904 SNP markers. The idea underlining the study was well conceptualized, the set objectives were measurable and achievable and the execution was appropriately carried out using adequate methodology. The results were well presented using good tables and figures.

Major shortcomings

Response

We appreciate your comments.

1. The justification of this study is too weak. What warrants comparing different genetic distance methods and clustering methods. If hierarchical clustering method is the most popular, what is responsible for that? What are the disadvantages of the other methods. I expect the authors to dig deeper into the nitty gritty of the methods to identify the knowledge gap more clearly.

Response

Thank you for your comment. We have added more information in the introduction to strengthen the justification for our study. This includes why hierarchical clustering is the most popular method for clustering, strengths and weakness etc. in the revised version of the manuscript as requested.

3. The comparison of the methods to identify the best method was not well presented. The authors claimed CCC was the basis but this is just a correlation coefficient, not a statistic to determine efficiency per se. More so, the CCC does not carry any test of significance in order to make inference.

Response

Thank you for the comment. We have used the CCC to determine the best heterotic grouping solution as it measures the correlation between the actual distances between the pairs of inbred lines and the cophenetic distance that represents the distance at which the inbreds are joined in the dendrogram. The CCC provides a measure of how well a hierarchical clustering tree represents the original distance matrix. Though CCC does not directly provide a level of significance. It however measures how well a hierarchical clustering tree represents the original distance matrix and as such used to compare different clustering methods or different distance metrics. A higher CCC indicates a better fit between the original distances and the dendrogram. Studies that have used CCC in similar manner include:

- Darkwa, K., et al. 2020. Comparative assessment of genetic diversity matrices and clustering methods in white Guinea yam (Dioscorea rotundata) based on morphological and molecular markers. Scientific reports, 10(1), p.13191.

- Bagheri L.M., et al. 2022. Screening for resistance and genetic population structure associated with Phytophthora capsici-pepper root and crown rot. Physiol. Mol. Plant Pathol.. 119, 101835. 10.1016/j.pmpp.2022.101924

Conclusion appears too wordy. It should be recast to capture the major findings based on the stated objectives.

Response

Thank you for the comment. We have recast the conclusion to capture major findings based on the stated objectives in the revised manuscript as suggested.

3. All references cited in the text were are listed in the reference section except Badu-Apraku et al., 2021

Response

We appreciate your comment. We have made the necessary corrections to the missing reference in the revised manuscript.

---

## [Editor Report · Decision Letter 1]

18 Dec 2024

Optimizing Breeding Strategies for Early-Maturing White Maize through Genetic Diversity and Population Structure

PONE-D-24-43431R1

Dear Dr. Adejumobi,

We’re pleased to inform you that your manuscript has been judged scientifically suitable for publication and will be formally accepted for publication once it meets all outstanding technical requirements.

Kind regards,

Muhammad Abdul Rehman Rashid, PhD

Academic Editor

PLOS ONE

---

## [Editor Report · Acceptance letter]

PONE-D-24-43431R1

PLOS ONE

Dear Dr. Adejumobi,

I'm pleased to inform you that your manuscript has been deemed suitable for publication in PLOS ONE. Congratulations! Your manuscript is now being handed over to our production team.

Kind regards,

on behalf of

Dr. Muhammad Abdul Rehman Rashid

Academic Editor

PLOS ONE